# A New Stitching Method for Dark-Field Surface Defects Inspection Based on Simplified Target-Tracking and Path Correction

**DOI:** 10.3390/s20020448

**Published:** 2020-01-13

**Authors:** Xue Chen, Jiaqi Li, Yongxin Sui

**Affiliations:** 1Changchun Institute of Optics, Fine Mechanics and Physics, Chinese Academy of Sciences, Changchun 130033, China; chenx@ciomp.ac.cn (X.C.); suiyongxin@cnepo.com.cn (Y.S.); 2Daheng College, University of Chinese Academy of Sciences, Beijing 100049, China; 3Changchun National Extreme Precision Optics Co., Ltd., Changchun 130033, China

**Keywords:** defect inspection, dark-field imaging, image stitching, target tracking, scanning path correction

## Abstract

A camera-based dark-field imaging system can effectively detect defects of microns on large optics by scanning and stitching sub-apertures with a small field of view. However, conventional stitching methods encounter problems of mismatches and location deviations, since few defects exist on the tested fine surface. In this paper, a highly efficient stitching method is proposed, based on a simplified target-tracking and adaptive scanning path correction. By increasing the number of sub-apertures and switching to camera perspective, the defects can be regarded as moving targets. A target-tracking procedure is firstly performed to obtain the marked targets. Then, the scanning path is corrected by minimizing the sum of deviations. The final stitching results are updated by re-using the target-tracking method. An experiment was carried out on an inspection of our specially designed testing sample. Subsequently, 118 defects were identified out of 120 truly existing defects, without stitching mismatches. The experiment results show that this method can help to reduce mismatches and location deviations of defects, and it was also effective in increasing the detectability for weak defects.

## 1. Introduction

Large-aperture precision optics are widely used in various fields, such as the National Ignition Facility (NIF), inertial confinement fusion system (ICF), ultra-high-power laser systems, shortwave optics, photo-lithography system, etc. [1,2,3,4,5]. Defects, such as scratches and digs on the surface of the optical element, will deteriorate the performance of the optic, especially in high-power laser and short-wave optical systems.

The camera-based dark-field imaging system has begun to be used for inspecting defects of large-aperture optical surfaces in recent years [6,7,8,9,10]. The tested sample surface is usually illuminated by annular-based light sources. The light scattered by the defects passes through a microscopic imaging system to form a dark-field image. Then, the images are captured by the camera with the characteristic of “dark background and bright targets”. Through sub-aperture scanning and stitching, the dark-field imaging system is able to inspect micrometer-level defects on large optical surfaces of tens of millimeters or even hundreds of millimeters over a relatively short period of time. Compared with other defect inspection methods, such as atomic force microscopy (AFM) and scanning electron microscopy (SEM), this camera-based inspection system presents a good balance between working performance and time consumed (especially for large optics) [11,12,13,14]. Compared with traditional human operators, the camera-based inspection system provides more repeatable and more reliable test results [15].

In order to obtain more detailed information of the surface defects, the field of view (FOV) of the inspection system should not be very large, usually designed to be several millimeters. Thus, for large optical surfaces, the sub-aperture scanning and stitching method is introduced into the inspection system to obtain the entire distribution of defects [6,7,9,12,13,14].

The design of the sub-aperture stitching method is one of the most challenging parts of this inspection system. Due to the existence of system errors, such as the translation stage positioning errors, the actual positions of each sub-aperture image may deviate from their nominal positions, which are calculated from the resolution of the camera, field of view (FOV), and stepping length of the translation stage, etc. If the stitching work is carried out simply with nominal positions, stitching failures and bad results might be obtained. For example, a single long run-through scratch can appear as two or more shorter scratches due to the dislocation of sub-apertures; a single dig located in overlapping areas can be judged as two individual digs.

To avoid those problems, several methods have been developed. The most widely used method is based on the “template matching” technique [6]. Defect features are taken into consideration to deal with the overlapping areas in adjacent sub-apertures. The “template matching” method fails when there are no defects or only one run-through line feature because it is difficult to find an accurate template. It is also quite time consuming in pixel-level template feature matching calculation. A detailed description of this problem has been discussed by Liu et al. in reference [14].

To solve the mismatch problems, Liu et al. proposed a feature-based multi-cycle stitching method, which was similar to a reasoning process [14]. This multi-cycle process has been proved to have good performance in avoiding mismatches for template-friendly features and mutual positioning condition. However, there are still some drawbacks in this method. On one hand, for discrete run-through line, there is not enough evidence for reasoning. On the other hand, system errors, such as translation stage positioning errors, will yield location deviation of defects, especially when there are no features in the overlapping area.

In this study, a stitching method based on simplified target-tracking and scanning path correction is proposed for the first time. By switching to the camera perspective (the camera is considered as a static observer, while the defects are treated as moving targets relative to the static observer) and increasing the sampling frequency (to make sure that each target appears at least three times), targets can be marked and initially stitched using the simplified target-tracking method. Scanning path correction is then executed by minimizing the sum of all marked targets’ deviations. The stitching results are updated by re-using the target-tracking-like method according to the corrected path. Finally, by eliminating the false alarms, the final stitching result can be obtained. Since the motion tracks are already known, it is relatively easy to track these moving targets when compared to common target-tracking applications like radar target-tracking [16,17,18,19].

There are mainly three advantages of this method:This stitching method is effective for reducing mismatches, even if there are only a few defects.This stitching method can make corrections on the scanning path deviation from the designed path. This scanning path deviation is usually one of the main sources resulting in location deviation of defects. Therefore, it is possible to reduce the location deviation of defects by eliminating the bad effects of the scanning path deviation.This stitching method has potential for improving the performance of weak defects detection while maintaining low false-alarm rates.

This paper is organized as follows: In Section 2, the schematic setup of our inspection system and the methods of sub-aperture scanning are presented; Section 3 gives a detailed description of our proposed sub-aperture stitching method. Comparison experiments between the conventional and our proposed stitching method are outlined in Section 4. Section 5 features discussion, and finally a conclusion to this paper is given in Section 6.

## 2. Dark-Field Microscopic Imaging System

### 2.1. System Layout

The schematic setup of the inspection system is shown in Figure 1. Annular spaced LEDs were used as the light source with a specific incidence angle onto the surface. As shown in Figure 2, if defects existed in the illuminated area, part of the incident light would be scattered into the microscopic imaging system (with camera and lens), which was placed on top of the tested surface. The defects would then be imaged by the imaging system, owing to the scattered light [20,21]. When no defects existed in the illuminated area, no incident light were reflected into the camera. Therefore, the output of the digital camera remained as dark images. This “dark-field” imaging system was able to achieve high sensitivity compared with other imaging methods [15,22].

The light source and camera were mounted onto a four-degree-of-freedom (4-DOF) translation stage to capture sub-aperture images. The 4-DOF translation stage made it possible to inspect both flat surfaces and sphere optical surfaces [23]. The object distance must be controlled precisely during the whole scanning period, to ensure the test surface within the depth-of-field of the imaging system.

Large surfaces with round contour were inspected by the built system. The scanning path for these surfaces was designed to be a set of concentric circles, whose centers coincided with the rotation center of the tested surface, as shown in Figure 3. As described in Section 1, the designed scanning path should ensure that each surface defect appears in at least three consecutive sub-apertures in both orthogonal directions. The occurrence time of each defect was determined by the overlapping area of adjacent sub-apertures. The area of overlapping zone was designed to be 2/3 of the sub-aperture.

### 2.2. Mismatch of Sub-Stitching

Because of the existence of system errors (such as translation stage positioning error), the full aperture image cannot be perfectly obtained according to the nominal positions. Sub-aperture stitching should be employed for compensating. However, even if template matching [6] or feature-based multi-cycle stitching method [14] are used, mismatches still exist. There are mainly two types of mismatches:Mismatch that increases the number of defects;Mismatch that decreases the number of defects.

#### 2.2.1. Mismatch That Increases the Number of Defects

For this kind of mismatch, one defect is judged as multiple defects. For example, a long run-through scratch is judged as two broken scratches, or a dig is considered as two. This kind of mismatch exists in both nominal position stitching and template-matching stitching with sampling order.

As shown in Figure 4a, there is a long run-through scratch S_1_ in the overlapping area. The actual position of the second frame is shown as a red solid line, while the nominal position for stitching is shown as a red dotted line. Due to the positioning error, there is a deviation between the two positions. After stitching, one scratch becomes two scratches, S_1_^(1)^ and S_1_^(2)^, as shown in Figure 4c. For digs, because of the existence of positioning error, one dig D_1_ in Figure 4b is distinguished as two digs, D_1_^(1)^ and D_1_^(2)^, after nominal position stitching, as shown in Figure 4d. For template-matching stitching with sampling order, the mismatch is detailed described in [14].

#### 2.2.2. Mismatch That Decreases the Number of Defects

For this kind of mismatch, two defects are judged as one defect. For example, two scratches are judged as one long run-through scratch, or two digs are considered as one dig. This kind of mismatch is common in template-based stitching, even for feature-based multi-cycle stitching.

As shown in Figure 5a, there is a scratch in the first frame and another scratch in the second frame. The angles of the two scratches are basically the same, but there is a certain distance between them. The actual position of the second frame is shown as a red solid, while the position for template-based stitching is shown as a red dotted line. Two scratches in the overlapping areas, S_1_ and S_2_, are treated as a long run-through scratch S’, as shown in Figure 5c. For digs, two different digs, D_1_ and D_2_ in Figure 5b in the overlapping area, are stitched into one single dig D’ in Figure 5d.

### 2.3. Location Deviation of Defects

Features of all suspected targets should be properly described quantitatively for preparation of the stitching work.

The dig can be featured by the following two factors:(1)The center coordinates of the target $\left(X_{c},Y_{c}\right)$;(2)The area of the target *S*.

Each scratch target is featured by its minimum bounding rectangles (MBR) (shown in Figure 6) with the following parameters:(1)The center coordinates $\left(X_{c},Y_{c}\right)$ of the MBR in the camera coordinate system, presenting the global location of the scratch;(2)The length and width of the MBR, *L* and *W*, presenting the size of the scratch;(3)The angle between length of MBR and *x* axis, θ, presenting the orientation of the scratch.

The square root of the difference between the actual position and the measurement result is defined as the location deviation Δl:(1)$$\Delta l=\sqrt{{\left(X_{c}-X_{c}{}^{\prime }\right)}^{2}+{\left(Y_{c}-Y_{c}{}^{\prime }\right)}^{2}},$$
where $\left(X_{c},\;Y_{c}\right)$ are the actual center coordinates of a dig or a scratch’s MBR of the actual position, and $\left(X_{c}{}^{\prime },\;Y_{c}{}^{\prime }\right)$ are the center coordinates of the measurement result.

If there are *n* defects on the surface, the mean location deviation (MLD) is designated as $\bar{\Delta l}$:(2)$$\bar{\Delta l}=\frac{\sum_{i=1}^{N}\Delta l_{i}}{N}.$$

The tested optics are usually of high quality and have only few defects, so most of the defects do not appear in the overlapping area. For the sub-apertures with no feature in the overlapping areas, sub-apertures are placed at the nominal positions. As is shown in Figure 7, there are two scratches ($S_{1}$ and $S_{2}$) in the second frame and no feature in the overlapping areas. The translation stage has positioning errors, and the measured result is $S_{1}{}^{\prime }$ and $S_{2}{}^{\prime }$. The system error, especially translation stage positioning error, yields a location deviation of defects.

## 3. Stitching Method for Dark-Field Surface Defects Inspection

The workflow of the proposed method is briefly described in Figure 8.

Image preprocessing should be firstly carried out, accomplishing target extraction from the original sub-apertures and the feature extraction of these suspected targets. Secondly, the initial stitching work should be performed, marking relevant suspected targets as the same defects. In this initial stitching work, the deviation tolerance of predicted position and actual position should be adequately larger due to the existence of both random positioning error of translation stage and the scanning path deviation. Thirdly, the scanning path compensation should be accomplished. Scanning path parameters were corrected according to the measurement of real scanning path accomplished by the inspection system. Then, the stitching work was updated according to the corrected scanning path parameters but with a smaller deviation tolerance between predicted position and actual position for each target. The deviation tolerance was estimated only by the random positioning errors of the translation stage. Finally, a process of target confirming was performed, eliminating false targets extracted in the former process.

In the following section we give a detailed description of the algorithm.

### 3.1. Basic Principle of the Simplified Target-Tracking Algorithm

As discussed above, each suspected target appeared in several adjacent sub-apertures under the designed scanning path. The static targets (defects, relative to the tested surface) appeared at different positions in adjacent sub-apertures. If considering the camera as a static observer, the suspected targets moved in certain motion tracks in a series of consecutive frames, which were completely determined by the sub-aperture scanning paths. The motion tracks could be precisely calculated by the scanning path parameters. Therefore, after a certain suspected target appeared in a certain sub-aperture for the first time, it was possible to precisely predict the appearance and positioning of the same target in the next two sub-apertures, under the condition that there was no positioning error.

If the tested surface was a curved surface, we firstly projected the surface as well as the scanning path onto a plane. Otherwise, if the tested surface was a plane surface, we took the plane itself as the reference plane. With the known scanning path settings, the nominal center coordinates of every sub-aperture $\left(x_{sa\_n},\;y_{sa\_n}\right)$ could be calculated in the global coordinate system, as well as the angle $\theta_{sa\_n}$ illustrating the angle between the *x* axis of camera coordinate and the *x* axis of the global coordinate. As in Figure 9, *n* is the serial number for each sub-aperture. The origin point of global coordinate system coincides with the rotation center of the tested surface.

After image preprocessing, a set of suspected targets were extracted from each sub-aperture. Now consider the case of a suspected target (surface defect) extracted from sub-aperture *n*. The coordinates of the suspected target are $\left(x{{}^{\prime }}_{tg\_n},\;y{{}^{\prime }}_{tg\_n}\right)$ in the camera coordinate system. As discussed above, the same target also appears in the next two sub-apertures, *n* + 1 and *n* + 2. Similarly, the coordinates of the target extracted from these two sub-apertures are defined as $\left(x{{}^{\prime }}_{tg\_n+1},\;y{{}^{\prime }}_{tg\_n+1}\right)$ and $\left(x{{}^{\prime }}_{tg\_n+2},\;y{{}^{\prime }}_{tg\_n+2}\right)$.

It is also possible to predict the coordinates of the same target in sub-apertures *n* + 1 and *n* + 2 using the sub-aperture *n* coordinates $\left(x{{}^{\prime }}_{tg\_n},\;y{{}^{\prime }}_{tg\_n}\right)$. Since the targets are stationary in the global coordinates, the predicting work can be carried out with a transformation between the camera coordinates of the two sub-apertures. In the following section we describe the mathematical deduction of predicted coordinates ($X_{n}^{n+1},\;Y_{n}^{n+1}$) in sub-aperture *n* + 1 and ($X_{n}^{n+2},\;Y_{n}^{n+2}$) in sub-aperture *n* + 2.

The coordinate transformation between two sub-aperture coordinates involves both translation and rotation. The translation between sub-apertures *n* and *n* + 1 can be expressed as:(3)$$\left\{\begin{array}{c} \Delta x_{sa}=x_{sa\_n+1}-x_{sa\_n}\\ \Delta y_{sa}=y_{sa\_n+1}-y_{sa\_n} \end{array}\right.$$
where ($x_{sa\_n}$, $y_{sa\_n}$) are the center coordinates of sub-aperture *n*, and ($x_{sa\_n+1}$, $y_{sa\_n+1}$) are the center coordinates of sub-aperture *n* + 1 in the global coordinates. They both can be calculated from the scanning path.

The rotation transformation between coordinates (x, y) and $\left(x^{\prime },\;y^{\prime }\right)$ by angle θ is:(4)$$\left\{\begin{array}{c} x^{\prime }=x\cos\theta -y\sin\theta \\ y^{\prime }=x\sin\theta +y\cos\theta \end{array}\right.$$

Here coordinates rotation between sub-apertures *n* and *n* + 1 is:(5)$$\theta =\;\theta_{sa\_n+1}-\theta_{sa\_n}$$

Now we are able to give the expression of the predicted coordinates as follows:(6)$$\left\{\begin{array}{c} X_{n}^{n+1}=x{{}^{\prime }}_{tg\_n}\cos\theta -y{{}^{\prime }}_{tg\_n}\sin\theta +\Delta x_{sa}\\ Y_{n}^{n+1}=x{{}^{\prime }}_{tg\_n}\sin\theta +y{{}^{\prime }}_{tg\_n}\cos\theta +\Delta y_{sa} \end{array}\right.$$
where $\left(x{{}^{\prime }}_{tg\_n},\;y{{}^{\prime }}_{tg\_n}\right)$ are the coordinates in sub-aperture *n*, θ is the coordinates rotation between sub-apertures *n*, and *n* + 1, ($X_{n}^{n+1},\;Y_{n}^{n+1}$) are the predicted coordinates of sub-aperture *n* + 1 according to $\left(x{{}^{\prime }}_{tg\_n},\;y{{}^{\prime }}_{tg\_n}\right)$.

The predicted coordinates in sub-aperture *n* + 2, ($X_{n}^{n+2},\;Y_{n}^{n+2}$) can be calculated with a similar method.

When there are no positioning errors, the coordinates of the corresponding target extracted from sub-apertures *n* + 1 and *n* + 2 should be the same with the predictive coordinates of this sub-aperture. Then we get the basic expressions in the target-tracking process:(7)$$\left\{\begin{array}{c} X_{n}^{n+1}={x^{\prime }}_{tg\_n+1}\\ Y_{n}^{n+1}={y^{\prime }}_{tg\_n+1} \end{array}\right.\;\left\{\begin{array}{c} X_{n}^{n+2}={x^{\prime }}_{tg\_n+2}\\ Y_{n}^{n+2}={y^{\prime }}_{tg\_n+2} \end{array}\right.$$
where $\left({x^{\prime }}_{tg_{n}+1},y{{}^{\prime }}_{tg\_n+1}\right)$ and $\left({x^{\prime }}_{tg_{n}+2},\;y{{}^{\prime }}_{tg\_n+2}\right)$ are the coordinates in sub-apertures *n* + 1 and *n* + 2, respectively.

### 3.2. Modified Feature-Based Target-Tracking-Like Stitching Method

The stitching method relies completely on the features of the surface defects. As described in Section 2.3, features of all suspected targets are properly described quantitatively in preparation of the stitching work. Digs, appearing in sub-apertures as point targets, can be simply described with their positions and diameters by calculating the center coordinates as well as the area of the marked connected pixel area on the image. Scratches, appearing in sub-apertures as line features, can be described by their minimum bounding rectangles (MBR), which are able to describe their lengths, widths, positions, and orientations. Here, we distinguished digs and scratches with the length/width ratio of their MBRs. Targets with MBR length/width ratio larger than 2.0 were marked as scratches, while the rest of them were marked as point targets including digs and dusts. This does not coincide with ISO standards [24] obviously. The criterion of judgment here was used only in the stitching work for better performance. Final inspection results should be given following the ISO standards or other standards, after the stitching work done.

For scratches, consider the stitch work of sub-apertures *n* and *n* + 1, where there is a run-through scratch. Using the target-tracking method described in Section 3.1, the scratch in sub-aperture *n* is transformed to the coordinates of sub-aperture *n* + 1, as shown with the black line in Figure 10a. The red line shown in the same figure is the extracted line feature in sub-aperture *n* + 1, which is part of the same run-through scratch. In the ideal case, part of these two lines should coincide completely with each other, and the two lines can be combined into a longer line feature. However, in practice, there are always system errors resulting in mismatch of these two line features. The green lines shown in Figure 10a illustrate the mismatch situation.

The maximum deviation of Δd and Δθ can be estimated with the distribution of system errors. If ∆*d* and ∆*θ* of a certain pairs of suspected targets exceed the maximum possible values of them, they should be marked as two separate defects. If both of these two values are within the possible regions, they should be marked as the same defects, in preparation for the next processing work. The rules that the transformation of suspected targets in sub-aperture *n* and the one in sub-aperture *n* + 1 can be marked as the same defect is that:(8)$$\left\{\begin{array}{c} \Delta d<T_{d}\\ \Delta \theta <T_{\theta } \end{array}\right.,$$
where $T_{d}$ and $T_{\theta }$ are the maximum deviations estimated with the distribution of current system error.

For digs, in existence of system errors, the predicted targets in sub-aperture *n* + 1 from sub-aperture *n* may also deviate from those extracted in sub-aperture *n* + 1. The extracted targets in sub-aperture *n* + 1 can appear in positions away from the position of same target in sub-aperture *n* with a distance ∆*d*, as shown in Figure 10b. The maximum deviation of ∆*d* can be estimated with the known system errors.

The rule that two suspected targets appearing in adjacent sub-apertures can be marked as the same target is expressed as:(9)$$\Delta d<T_{d},$$
where $T_{d}$ is the maximum deviation estimated with the distribution of current system error.

### 3.3. Scanning Path Correction

The challenges faced by the stitching work mostly arise from the difference between nominal position and actual position of each sub-aperture, which is difficult to predict or estimate. There are mainly two sources of this position offset. The first one is the random positioning error of the translation stage, which follows the Gaussian distribution. The other one is the departure of the scanning path away from the ideal scanning path. The former error source is stable and cannot be eliminated, while the latter is possible to be corrected.

As per the description in Section 2, the scanning path of our system is a set of concentric circles. In the ideal case, the center should coincide with the center of rotation of the tested surface. Based on our experience, there are mainly two kinds of scanning path departure for the system:Under the situation where the position of the starting point of the scanning has a deviation apart from the ideal position, which often happened in our practical works, the actual path scanning by the system is another set of concentric circles, as shown by the dashed lines in Figure 11a;The centers of scanning circles do not coincide with the rotation center of the tested surface, as shown in Figure 11b. Compared with the ideal scanning path, the actual position of every sub-aperture departs from its nominal position.

It is possible to correct these kinds of departures of scanning path. Suppose there are *n* targets extracted after the stitching method. As has been discussed above, each target appeared in least three consecutive sub-apertures. After a certain target appears for the first time, it is possible to predict its appearance in the next (second) and the third sub-aperture. The deviation between the predicted appearance and the actual appearance in the second and third sub-aperture arises partly from the departure of scanning path. Thus it is possible to evaluate the level of scanning path departure through the value of deviation. In a mathematical expression, calculate the deviation of actual coordinates ($x_{a},\;y_{a}$) and predicted coordinates ($x_{p},\;y_{p}$) in the second and the third sub-aperture for each target. Take target *n* as example, the deviation $D_{n}$ is:(10)$$D_{n}={\left\{{\left(x_{a}-x_{p}\right)}^{2}+{\left(y_{a}-y_{p}\right)}^{2}\right\}}_{2nd}+{\left\{{\left(x_{a}-x_{p}\right)}^{2}+{\left(y_{a}-y_{p}\right)}^{2}\right\}}_{3rd}.$$

The predicted coordinates ($x_{p},\;y_{p}$) are calculated from specific scanning path parameters. If these parameters largely deviate from the actual scanning path, the predicted coordinates of targets also largely deviate from the real positions where targets appear on sub-apertures, as shown in Figure 12. On the contrary, if the scanning path parameters used for stitching sub-apertures are close to real scanning parameters, the deviation $D_{n}$ tends to be negligible. Therefore, it is possible to correct the scanning path parameters by making efforts to minimize the sum of all targets’ deviations, which can be expressed as:(11)$$\sum_{n=1}^{N}D_{n}\to 0.$$

The corrected scanning path parameters result in more precise positions of sub-apertures, thus achieving better stitching performance.

### 3.4. Potential Enhancement of Weak Defect Detection through Target Reconfirming

In Section 2, we introduce that the inspection system detects defects of tested surfaces by collecting their scattered light. Large defects on surfaces with large scattering cross section scatter the illumination light strongly, making it easier to be captured by the detector (camera). However, there are always some weak defects, which might be narrow and shallow, scattering insufficient light into the camera. As a result, the camera responds weakly against these weak defects, as shown by the little bulge in the red circle in Figure 13.

This system involves a threshold-segmentation method extracting suspected targets from original sub-apertures [25]. The threshold should be carefully chosen to achieve a good balance between detecting performance and false-alarm rate. The weak defects can be better extracted when a lower threshold is applied to the image preprocessing. However, the low threshold can introduce background noise and uneven illumination areas to be judged as real defects on the tested surface, leading to bad results of the inspection work. For the existing methods, the performance in detecting weak defects, especially digs, is always sacrificed to some extent to avoid bad inspection results.

Our proposed method has potential for helping to improve the performance of weak defects detection while maintaining low false-alarm rates. As discussed above, each real defect appears in at least three consecutive sub-apertures, with a motion track coinciding with sub-aperture scanning path. False targets, however, appear at random positions in each sub-aperture. Therefore, we are able to make a distinction between real targets and false alarms by the occurrence number of each targets. As shown in Figure 14, red spots, green spots, and blue spots are the transform projections of suspected targets from sub-apertures *n*, *n* + 1, and *n* + 2 respectively. The targets 1, 2, 3 appear as a motion track in the field of view, while the other three targets appear at random positions. We are able to make the judgments according to this appearance. The targets with occurrence less than 3 are treated as false alarms and are eliminated.

Because this method can effectively reduce false alarms, the threshold that is used for extracting suspected targets from original sub-apertures can be appropriately decreased in the image preprocessing stage. Thus, the weak target detection ability of the system is indirectly improved.

By properly setting the threshold in the preparation stage and eliminating false targets in the target confirming cycle, our proposed method can improve the performance of weak defects detection while maintaining low false-alarm rates.

## 4. Experiment

### 4.1. Experiment System

The experiment system is shown in Figure 15 and described in Section 2.1. The dark-field defect inspection system mainly included 4-DOF translation stage, annular spaced LED light source, and microscopic imaging system. The capture speed of the camera can reach 75 frames per second with image size of 2048 × 2048 in pixels. The magnification of the Edmund telecentric lens was 1.7× and the depth of field was ±0.18 mm at f/10. The field of view of the microscopic imaging system was 3 × 3 mm. The light source was a 460 nm annular-based LED with the incident angle of 45°. The accuracy of measurement on size and position was about 1.5 microns.

### 4.2. Defect Distribution Map for Sample Surface

To experimentally evaluate the performance of our proposed method, a sphere sample surface (φ 120 mm) was inspected by the system. The image shown in Figure 16a is the combination of original sub-apertures, which is composed of approximately ten thousands of 3 × 3 mm sub-apertures with a large overlapping area as described earlier. A series of steps, including image preprocessing, initial stitching work, scanning path correction, updating stitching work, and target confirmation were applied to work out the final extraction of surface defects. As shown in Figure 16b, the extracted defects were drawn down one by one with their real sizes and positioning, and the pink area illustrates the tested round surface. Images like Figure 16b are named as “distribution maps” in the following. In the inspection result of the tested surface, several long scratches and a great number of point defects (including digs and dust particles) are shown in the distribution map. Compared with original sub-apertures, the distribution map offers better visibility of defects, especially for small and weak defects.

### 4.3. Stitching Performance against Sub-Aperture Mismatches

The stitching and scanning path correction method described in Section 3 was performed to deal with the sub-aperture mismatches. The mismatches of both digs and scratches could be corrected, as illustrated in the following cases.

The defect distribution map shown in Figure 17a is the central part of the original stitching result of the tested surface. All defects shown in the distribution map are point defects (including digs and dusts). It might be confusing that most of these defects look like worms, rather than points or small circles. In fact, the worm-like appearances are caused mainly by the deviation of sub-aperture scanning path from the designed path, whose parameters were directly used for the stitching work. The deviated scanning path imposed a cumulative deviation between the transformed coordinates of a certain defect in consecutive sub-apertures. The cumulative deviation of several consecutive sub-apertures finally led to the worm-like stitching result.

As discussed in Section 3, the deviated scanning path could be corrected by minimizing the overall deviation of all suspected targets, which is expressed by Equation (11). The worm-like stitching result was fixed with scanning path correction process. The residual effects of mismatches caused by random positioning errors could be suppressed by the simplified target-tracking stitching algorithm. The final extraction result of these defects is shown in Figure 17b, presenting the number, sizes, and positions of these defects with high reliability. The detailed original distribution map and corrected distribution map for area “A” are shown in Figure 17c,d.

A long run-through scratch mismatch correction work is described as the second case. The following three images shown in Figure 18 illustrate three different situations inspecting a certain surface scratch. For the better description and visual presentation of this case, both original images and the feature MBR of scratches are present. The image in Figure 18a is composed of two adjacent sub-apertures, with a small overlapping area in accordance with the scanning strategy in conventional stitching methods. A serious mismatch can found at position “A” in the middle of this scratch. Increasing the overlapping area as required by our proposed method (at least three appearances for each target), the same scratch is composed of four sub-apertures in Figure 18b. Mismatches still exist in overlapping areas “B”, “C”, and “D” but are smaller than that of Figure 18a. The mismatch became smaller because of the smaller cumulative deviation between adjacent sub-apertures, as space between sub-apertures decreased.

The stitching result was finally improved by our stitching method, as shown in Figure 18c. The image is composed of four sub-apertures, the same as those in Figure 18b. The mismatches are almost invisible, making it much easier to be properly stitched.

In fact, all of these images can stitched properly under different rule settings described in Section 3.2. Appropriate threshold values should be set to make proper combination of mismatched scratches. Large threshold values were used to deal with more serious mismatches. It should be noted that small threshold values are always more desirable, because larger values might cause another type of stitching failure in which two real scratches close to each other are combined into one single scratch. For the raw data acquired with our proposed scanning strategy, especially after scanning path correction, smaller threshold settings could be used, which were beneficial for better stitching results.

### 4.4. Contrast Experiment

Contrast experiments on our stitching method are presented in this section, with a comparison to nominal position stitching (NPS), template match stitching (TMS) [6], and feature-based multi-cycle stitching (MCS) [14]. A special designed test sample was used to calibrate the real size of the defects. This was a round fused quartz plate with a diameter of 120 mm. A total of 120 defects (60 digs and 60 scratches) were grooved at the planned position on the test sample, and 10 of them were weak defects with shallow depth (below 50 nm [22]). The sub-apertures were 2048 × 2048 pixels and the field of view (FOV) of the microscope was 3 × 3 mm. The positioning error of each axis on the translation stage used for sub-apertures sampling was ±10 μm. The full aperture image was obtained by stitching the sub-apertures together.

The comparison results of mismatches, MLD, recognition rate, scanning time, and data processing time are shown in Table 1. The same hardware was used in the contrast experiment (C# with Halcon imaging-process library, 64 bit Windows 7 operating system, Intel Core i5 processor, 4 GB DDR3 1333 MHz memory). The settings for overlapping areas of the sub-apertures were a little different. The area overlapping ratio was set to 2/3 in our scanning path design, while 1/6 in NPS, TMS, and MCS. The results of the contrast experiment are shown in Table 1.

Experiments showed that this method could effectively reduce the number of mismatches. By path correction, the defect positions in the test results were more accurate than the other methods with a minimum MLD value of 10.67 μm. In addition, the target-tracking stitching method could also improve the detection ability of defects.

## 5. Discussion

The starting point of the proposed method is to obtain more reliable stitching results by utilizing sufficient data and information. It should be noticed that the challenges faced by the conventional methods basically arise from insufficient information. Conventional methods have to introduce algorithms with more complexity to achieve the ability to make accurate analyses and inferences with insufficient information. As for our proposed method, the scanning strategy was modified in comparison with the conventional methods. The scanning process acquires more sub-apertures, which means that mass data and more sufficient information can be used by the algorithm. The mass data and sufficient information make it possible for the algorithm to make identifications of the targets through their moving tracks, make compensations on scanning path parameters by minimizing the predicted values and actually extracted values of targets, and then obtain reliable and precise stitching results.

The main drawback of our proposed method is that it requires more sub-apertures than existing methods. To ensure that the whole surface is covered in existence of the sub-aperture positioning errors, a small overlapping area between two adjacent sub-apertures is enough for existing methods. However, in the proposed method, the target-tracking algorithm requires every single target appearing in at least three adjacent sub-apertures for each direction (*x* and *y* axis), which means the number of sub-apertures increases by about nine times. We notice that more sub-apertures might be more time consuming in both the scanning procedure and image processing procedure. This drawback, however, could be avoided to some extent based on our experience. The time for scanning could be reduced by utilizing cameras with higher frame rates and larger sensor areas. The time consumed in the image processing is at the same level with respect to reference [14] due to the simpler algorithm and non-pixel-level operation, even if there are more sub-apertures. The simplification of our proposed method arises from a different perspective on the original sub-apertures. The methods in this paper focus completely on the targets themselves, while the conventional methods treat mostly the sub-apertures. It should be noted that little effective information exists on the original sub-apertures; thus, most of the calculation work is meaningless, resulting in a lengthier process. The calculation and time resources were best utilized by focusing only on targets, resulting in less time expenditure in this method.

## 6. Conclusions

The camera-based dark-field imaging system is an effective way to evaluate surface defects. However, it is challenging to inspect the micron-sized defects, which need to be positioned and quantified accurately over the whole large fine optical surfaces of hundreds of millimeters. Since there are few defects on the tested fine optics, especially in the overlapping areas, the conventional stitching methods face problems of mismatching and location deviation. A new simplified stitching method based on target-tracking and adaptive scanning path correction is proposed in this paper. Instead of focusing on sub-apertures, this method focuses completely on the targets themselves. By increasing the number of sub-apertures and changing the camera perspective, the defects can be treated as moving targets. After image preprocessing, a target-tracking-like procedure is firstly carried out to get the marked targets. Then, by minimizing the sum of all marked targets’ deviations, scanning path can be corrected. The final stitching results are updated by re-using the target-tracking-like method according to the corrected path and eliminating false alarms. Experiments show that the proposed method has good performance in avoiding mismatches and decreasing location deviations of defects. Meanwhile, it has potential for helping to improve the performance of weak defects detection while maintaining low false-alarm rates. This stitching method has been applied in the defects inspection of photolithography lens and has achieved good results.

## Figures and Tables

**Figure 1 sensors-20-00448-f001:**
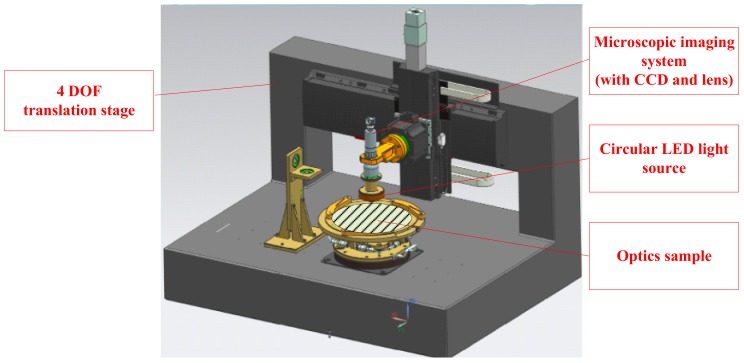
Schematic setup of the inspection system.

**Figure 2 sensors-20-00448-f002:**
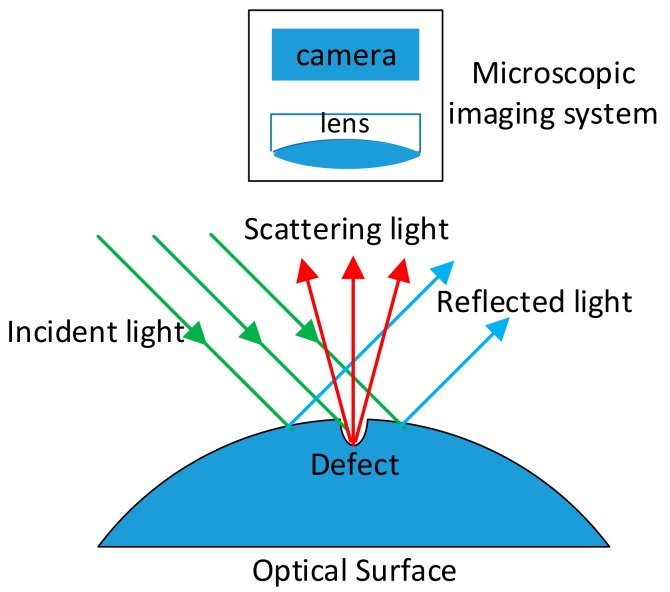
Principle of dark-field imaging system.

**Figure 3 sensors-20-00448-f003:**
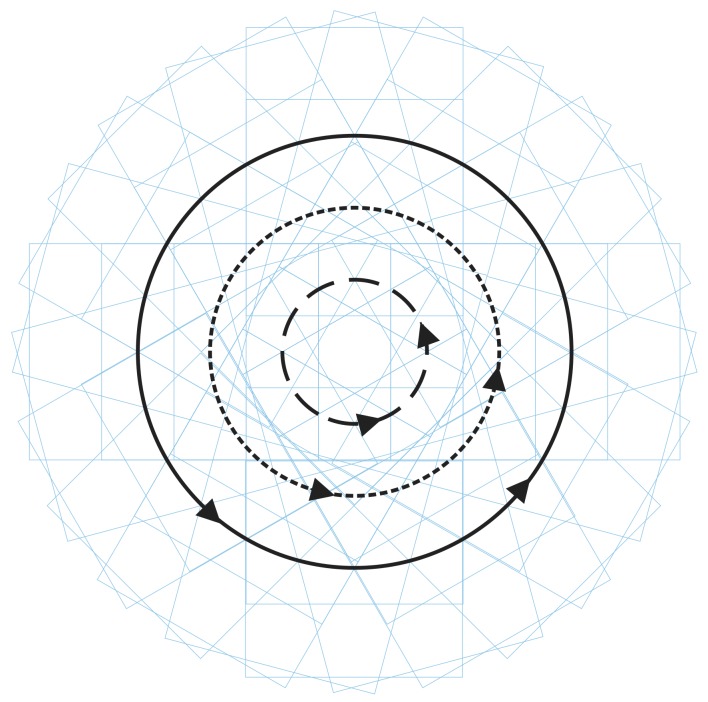
Scanning path of large surface with round contour.

**Figure 4 sensors-20-00448-f004:**
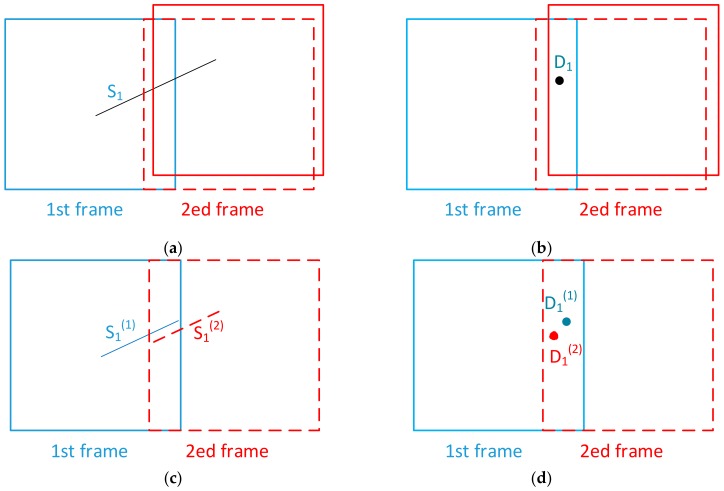
Mismatches that increase the number of defects. (**a**) Actual scratch; (**b**) Actual dig; (**c**) Mismatch for scratch; (**d**) Mismatch for dig.

**Figure 5 sensors-20-00448-f005:**
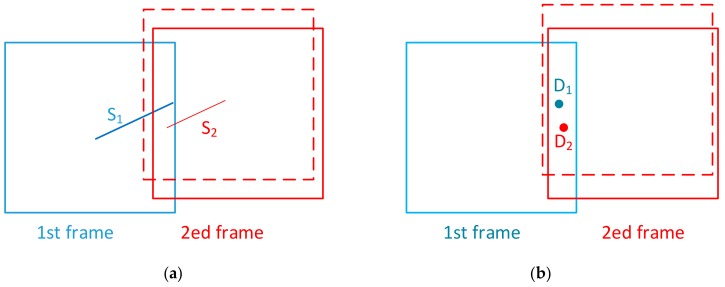
Mismatches that decrease the number of defects. (**a**) Actual scratch; (**b**) Actual dig; (**c**) Mismatch for scratch; (**d**) Mismatch for dig.

**Figure 6 sensors-20-00448-f006:**
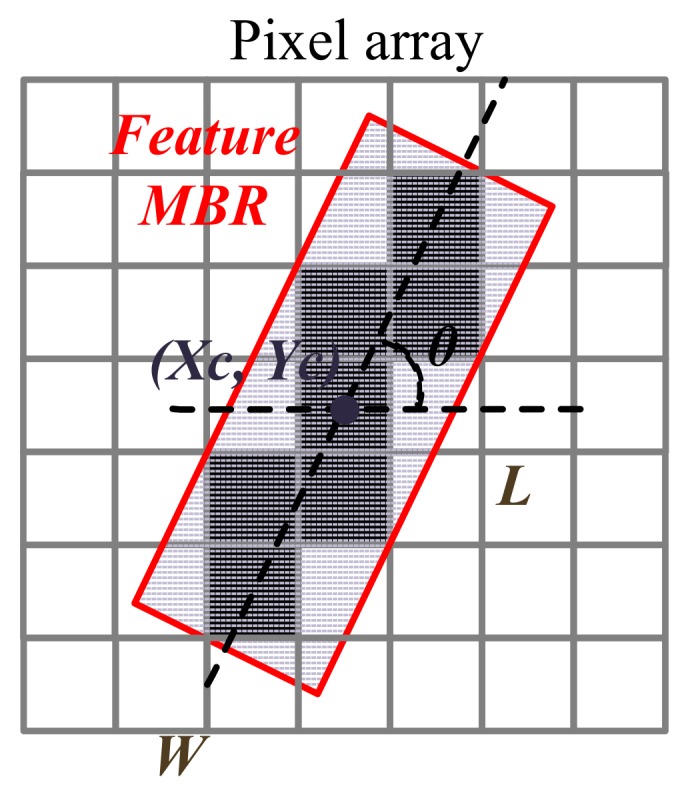
Minimum bounding rectangles (MBR) of a scratch.

**Figure 7 sensors-20-00448-f007:**
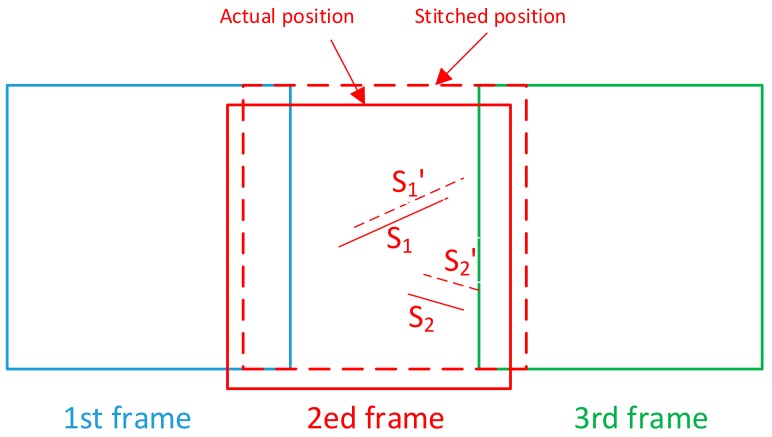
Location deviation of defects caused by sub-apertures with no feature in the overlapping area by the conventional method.

**Figure 8 sensors-20-00448-f008:**
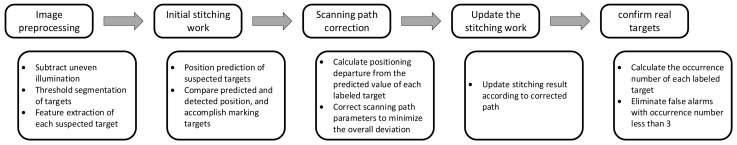
Diagram of the workflow.

**Figure 9 sensors-20-00448-f009:**
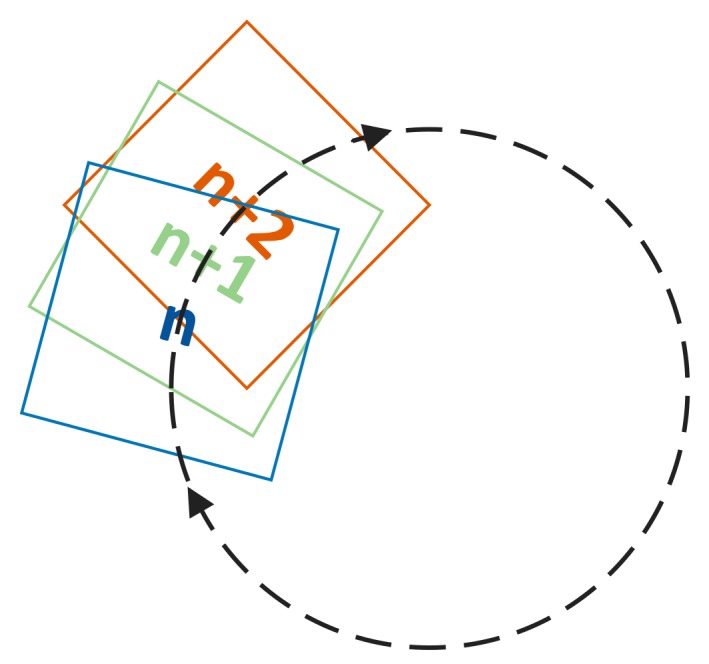
Scanning path and number rules during scanning and data processing.

**Figure 10 sensors-20-00448-f010:**
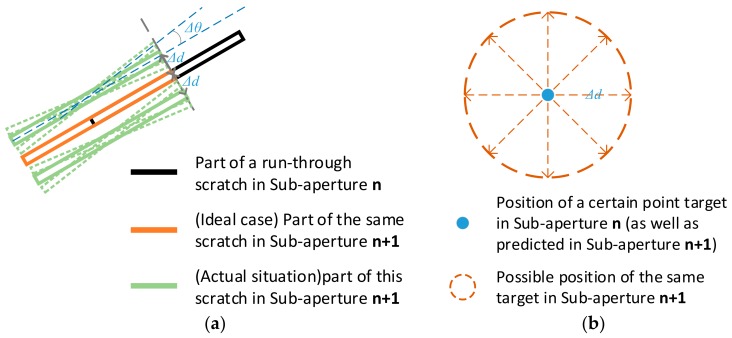
Description of mismatches for scratch and digs situation. (**a**) Mismatch of long run-through scratches between consecutive sub-apertures; (**b**) Possible dislocation of the same point targets between consecutive sub-apertures.

**Figure 11 sensors-20-00448-f011:**
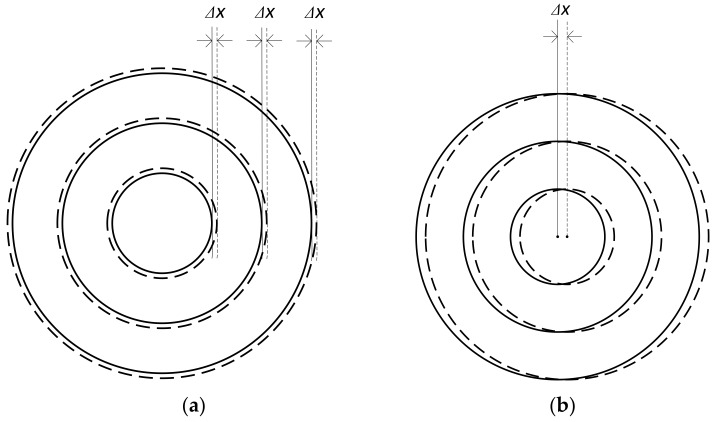
Types of scanning errors. (**a**) The first kind of scanning error, homogeneous radius error caused by the deviation of the starting position; (**b**) The second kind of scanning error, center shift between the planned and actual scanning path.

**Figure 12 sensors-20-00448-f012:**
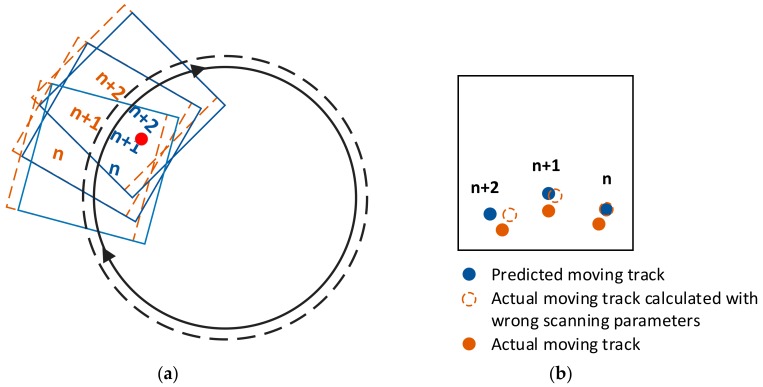
(**a**) Comparison between sub-apertures under ideal scanning settings and actual scanning path; (**b**) Description of how scanning path departures affect the stitching work.

**Figure 13 sensors-20-00448-f013:**
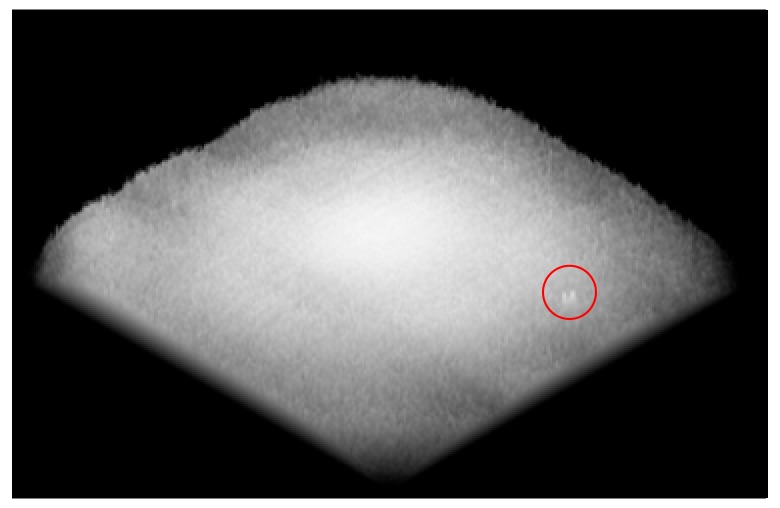
A 3D presentation (2D sensor pixel array—gray level) of the camera response to a weak defect on the tested surface. The height at each pixel represents for the gray value of the camera sensor response.

**Figure 14 sensors-20-00448-f014:**
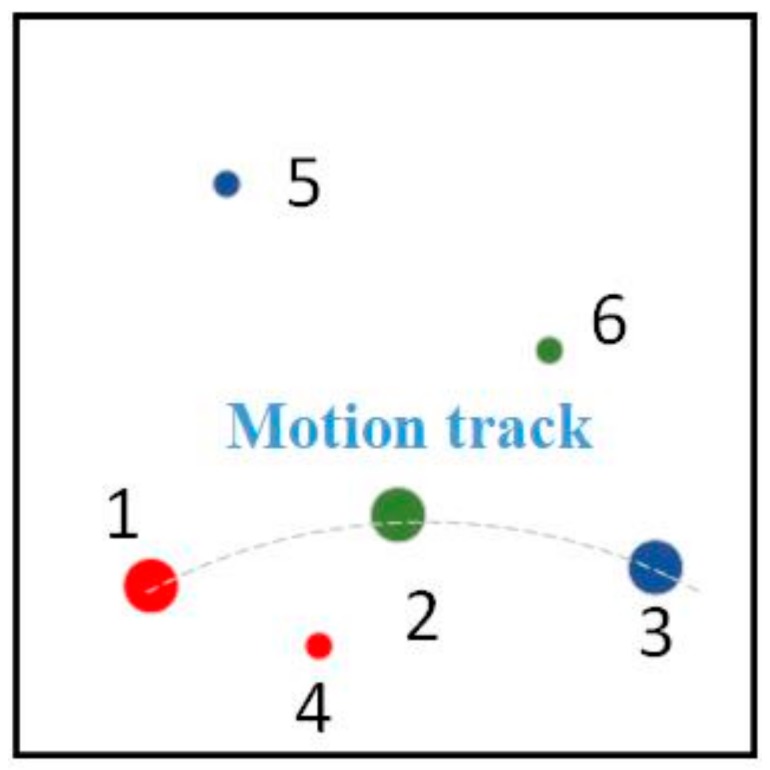
Superimposition of consecutive three sub-apertures (or the three consecutive frames of image from the camera view).

**Figure 15 sensors-20-00448-f015:**
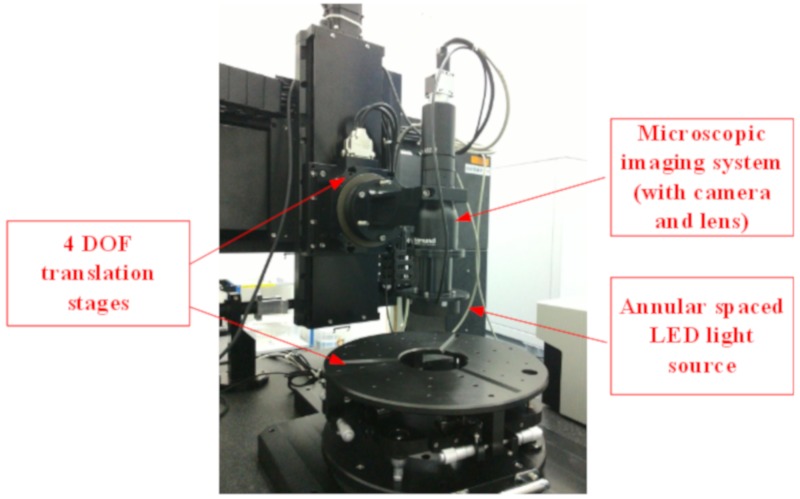
Dark-field defect inspection system with four-degree-of-freedom (4-DOF) translation stage, annular spaced LED light source and microscopic imaging system.

**Figure 16 sensors-20-00448-f016:**
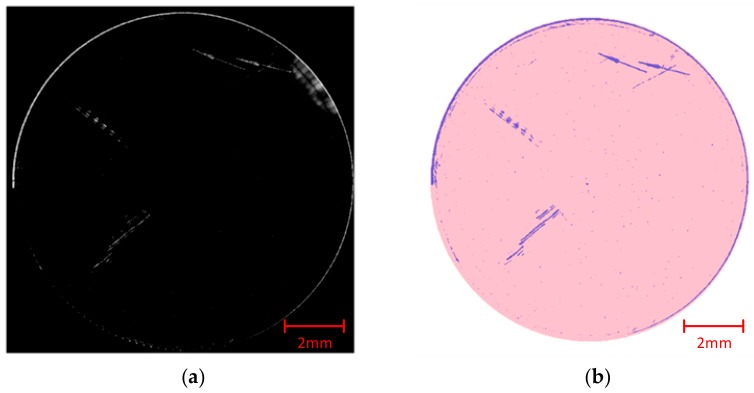
(**a**) The stitching result of original images on inspection of a tested surface; (**b**) The distribution map of extracted defects on the tested surface.

**Figure 17 sensors-20-00448-f017:**
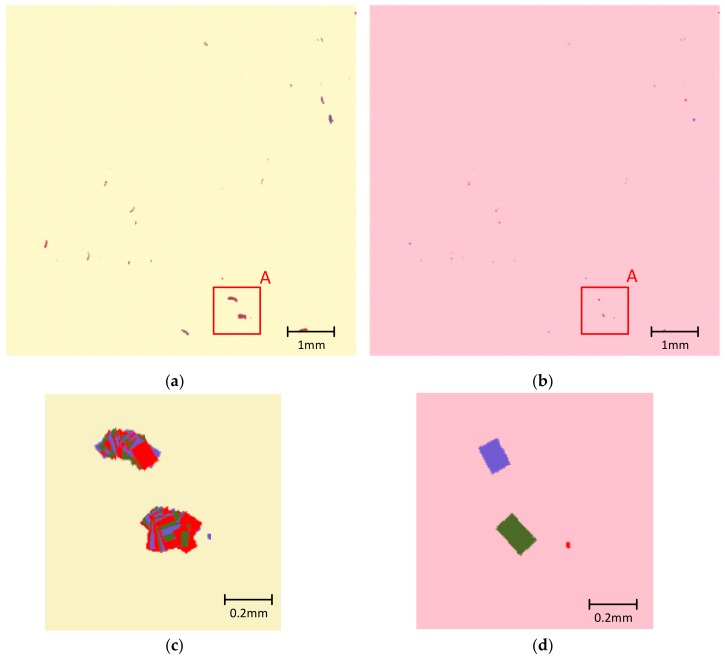
Comparison between stitching results of point targets before and after the scanning path correction work. (**a**) Original distribution map of point defects on the tested surface, with worm-like defect distribution in existence of system errors; (**b**) The corrected distribution map after scanning path correction process; (**c**) Detailed view of part A in the original distribution map; (**d**) Detailed view of part A in the corrected distribution map.

**Figure 18 sensors-20-00448-f018:**
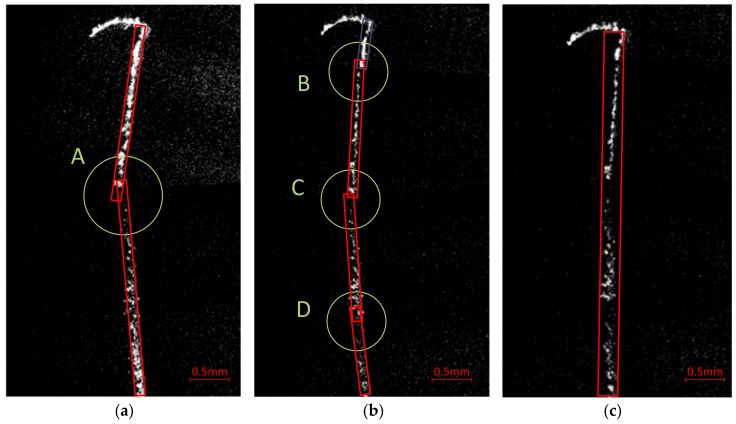
Improvement of scratch mismatch correction with and without our proposed method. (**a**) Stitching result with conventional stitching strategy (two sub-apertures with small overlapping area); (**b**) Initial stitching result with four consecutive sub-apertures, in accordance with our scanning and stitching strategy; (**c**) Stitching result after the scanning path correction process.

**Table 1 sensors-20-00448-t001:** Results comparison.

Stitching Method	Nominal Position Stitching	Template Match Stitching	Multi-Cycle Stitching	Our Method
Mismatches of sub-stitching	25	8	2	0
Mean location deviation (μm)	68.46	27.86	18.52	10.67
The number of defects identified	110	110	110	118

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
