# Peer review of "A New Stitching Method for Dark-Field Surface Defects Inspection Based on Simplified Target-Tracking and Path Correction"

_sensors, 2020, doi:10.3390/s20020448_

Round 1

Reviewer 1 Report

The method presented in the article is a new method. There is no doubt about it. The authors made a number of mistakes and inaccuracies in the article. 

In short, this is an article on the subject - image processing.

The authors should explain exactly how the exact measurement of defects on the test surface was carried out.  What is the accuracy of this measurement? What accuracy does the author's method provide? 

The authors in table 1 give the number 10.67 microns - this means that the method allows you to measure with an error of 0.01 microns? What does "shallow depth" mean? How much exactly? What is the depth of other defects? 

On a number of images it is necessary to put a scale (Fig. 16, 17,18). 

Too General information in Abstract - there is not a single number.

Reviewer 2 Report

The paper is well written and well organized.

I have here to remark that in figure 11(a) different radii should be denoted with different Δx, currently they seem to be equal.

Another critical point regards the indices notation of equations (3-7). I am not sure that is the best and more clear choice.

The experimental setup should be described in detail also in the caption of fig. (15).

Essentially, I suggest its publication.
